# Link between Lipid Second Messengers and Osmotic Stress in Plants

**DOI:** 10.3390/ijms22052658

**Published:** 2021-03-06

**Authors:** Beatriz A. Rodas-Junco, Graciela E. Racagni-Di-Palma, Michel Canul-Chan, Javier Usorach, S. M. Teresa Hernández-Sotomayor

**Affiliations:** 1CONACYT—Facultad de Ingeniería Química, Campus de Ciencias Exactas e Ingenierías, Universidad Autónoma de Yucatán (UADY), Periférico Norte Kilómetro 33.5, Tablaje Catastral 13615 Chuburná de Hidalgo Inn, C.P. 97203 Mérida, Mexico; 2Departamento de Biología Molecular, Universidad Nacional de Río Cuarto, C.P. 5800 Río Cuarto, Argentina; g_ragacni@hotmail.com; 3Facultad de Ciencias Químicas, Universidad Veracruzana, Prolongación de Avenida Oriente 6 Num. 1009, Rafael Alvarado, C.P. 94340 Orizaba, Mexico; mcanul@uv.mx; 4Unidad de Bioquímica y Biología Molecular de Plantas, Centro de Investigación Científica de Yucatán (CICY), Calle 43 No. 130, Col. Chuburná de Hidalgo, C.P. 97205 Mérida, Mexico; jiusorach@unvime.edu.ar

**Keywords:** lipid messengers, phosphatidic acid, phospholipase C, phospholipase D, sphingholipids, lysophospholipids

## Abstract

Plants are subject to different types of stress, which consequently affect their growth and development. They have developed mechanisms for recognizing and processing an extracellular signal. Second messengers are transient molecules that modulate the physiological responses in plant cells under stress conditions. In this sense, it has been shown in various plant models that membrane lipids are substrates for the generation of second lipid messengers such as phosphoinositide, phosphatidic acid, sphingolipids, and lysophospholipids. In recent years, research on lipid second messengers has been moving toward using genetic and molecular approaches to reveal the molecular setting in which these molecules act in response to osmotic stress. In this sense, these studies have established that second messengers can transiently recruit target proteins to the membrane and, therefore, affect protein conformation, activity, and gene expression. This review summarizes recent advances in responses related to the link between lipid second messengers and osmotic stress in plant cells.

## 1. Introduction

Plants use complex signal transduction networks to orchestrate biochemical, genetic, and physiological responses under different stress conditions. Among the components involved in that response are molecules called second messengers. These molecules are “master regulators” since they generate a high degree of amplification via signal transduction and modulate key downstream molecular regulatory components involved in the response to stress. Lipids are major components of biological membranes that serve as platforms for important signaling functions [1,2]. Lipid-second messengers may be formed from membrane structural lipids by hydrolytic activity of phospholipases such as phospholipase D (PLD), phospholipase C (PLC), and phospholipase A2 (PLA2). In the context of stress in plants, salt, and drought represent osmotic factors that limit crop productivity [3]. In this context, salt or drought are different types of stress that result in a series of different changes at the cellular or plant level, generating specific changes at the biochemical, molecular, and physiological levels in plants. Understanding the molecular mechanism by which plants respond to osmotic stress signals is pivotal for the development of biotechnological tools for the generation of tolerant plants. This review will focus on assessing the current knowledge of lipids second messengers (phosphoinositides, phosphatidic acid, sphingolipids, and lysophospholipids), which have been shown to be key regulators of osmotic stress responses in plant cells.

## 2. Lipid-Derived Second Messengers in Plant Cells

Phospholipids are important components in all membranes in eukaryotes and play a role in signaling mechanisms in plant cells. Enzymes as phospholipases, lipid kinases or phosphatases modify membrane lipids to generate signaling molecules known as lipid-derived second messengers. Important lipid second messengers include phosphatidylinositols, diacylglycerols, phosphatidic acid, sphingolipids, and lysophospholipids [4,5] (Figure 1). Several research groups have reported that lipid second messengers activate or recruit proteins to membranes, which leads to the activation of downstream signaling pathways that result in cellular events and physiological responses. In this review, we will attempt to highlight some of the recent studies on the of the functional mechanism of lipid-derived second messengers, with an emphasis on their regulation, particularly in response to osmotic in plant cells.

## 3. Phosphoinositide Signaling

Phosphoinositides (PI) are a class of inositol-containing phospholipids present in the plasma membrane. In plants, the inositol ring is sequentially phosphorylated at several different positions, generating five isomers: phosphatidylinositol (PI), PI-3 phosphate (PI3P), PI-4 phosphate (PI4P or PIP), PI-5 phosphate, PI-3-5- bisphosphate (PI-3,5-P_2_), and PI-4,5-bisphosphate (PI-4,5-P_2_ or PIP_2_) [6,7].

Unlike the majority of membrane lipids, PIs show only a minor abundance, and their dynamic formation occurs a set of specific kinases and phosphatases, and is maintained via constant turnover [8]. Additionally, they modulate fundamental cellular processes, such as membrane trafficking, cytoskeleton organization, polar tip growth, and stress responses [9]. At the poles across kingdoms, phosphoinositide is involved in polar tip growth [10]. PIs can work as ligands for different proteins called PI “modulins” and regulate their subcellular distribution or activity via interactions. PI binding takes place through the inositol polyphosphate head groups and PI binding domains of phosphoinositide, such as pleckstrin homology (PH) domains, Phox homology (POX) domains, and Fab1-YOTB-Vac1-EEA1 (FYVE) domains [8]. Examples of PI modulin activities include the regulation of ion channels [9], ATPase activity, and hormonal and stress signaling [9]. In Arabidopsis and rice, the presence of proteins with FVYE domain has been reported in response to abiotic stress tolerance [11]. In phosphoinositide signaling, the generation of a second messenger occurs through the activation of phospholipases. PI-phospholipase C (PLC) catalyzes the hydrolysis of PIP_2_ to generate the soluble second messenger’s inositol 1,4,5-trisphosphate (IP_3_) and diacylglycerol (DAG). In plants, DAG is converted into phosphatidic acid (PA), while IP_3_ may be further phosphorylated to form inositol hexakisphosphate (IP_6_). PA may also be generated by hydrolysis of structural phospholipids such as phosphatidylcholine (PC) and phosphatidylethanolamine (PE) by phospholipase D (PLD) enzymes [12]. Although the study of these lipid second messengers has provided evidence of their importance in plant defense response under stress, many questions still need to be answered. As a continuation, the roles of IP_3_, IP_6_, PA, and other lipid second messengers in plants are described below.

### 3.1. IP_3_ as a Second Messenger in Plant Cells

PIs constitute a class of membrane phospholipids that are substrates for phosphoinositide-specific phospholipase C (PI-PLC). PI-PLC catalyzes the hydrolysis of PIP_2_ to generate two important second messengers, IP_3_ and DAG [12]. In plant systems, the role of IP_3_ in releasing Ca^2+^ from cellular stores has been widely reported [13]. However, a critical component that is still unknown in plant cells is a putative IP_3_ receptor (IP_3_-R). The search for an IP_3_-R has been underway for many years. Various authors have approached the search for an IP_3_ receptor through in silico and in vivo analyses, and an interesting approach that has been taken is the search for homologous gene(s) that encode the IP_3_ receptor in plants. Sequencing of the green algae Chlamydomonas sp. genome, which does possess such a receptor, has made it possible to generate valuable genetic information to explain that this gene has been discarded during the evolution of plants. Additionally, at the protein level this does not clarify whether plants express an IP_3-_R, as it indicates only that there is no plant protein that has an IP_3_ receptor RIH domain [ryanodine, (RYR) and IP_3_ homology] in animals in structural homolog databases [14,15].

On the other hand, an interesting aspect of IP_3_ as a second messenger that is well documented is the rapid intracellular changes that this molecule shows under biotic or abiotic stimulation. For example, Monteiro et al. [16] reported that IP_3_ caused an influx of Ca^2+^ in the growing pollen tube of *Agapanthus umbellatus* under osmotic shock treatment. Additionally, biphasic changes in IP_3_ were detected in response to gravity in *Arabidopsis* inflorescence stems [17] and *Avena sativa* [18] or cold exposure in Arabidopsis suspension cells [19]. The release of IP_3_ has often been linked to the activation of PI-PLC [13,14,20]. For example, in Arabidopsis, an increase in IP_3_ via PI-PLC activation in response to blue light induces the release of Ca^2+^ [21]. Legendre et al. [22] hypothesize that the activation of PI-PLC and increase in IP_3_ could be a way by which polygalacturonic acid triggers an oxidative burst in soybean cell suspensions. Recently, Ren et al. [23] showed that the increase in IP_3_ after heat shock in Arabidopsis plants is partially dependent on the activity of AtPLC3 (Arabidopsis thaliana Phosphoinositide-Specific Phospholipase C Isoform 3). Collectively, these examples indicate that the increase in IP_3_ as a consequence of PI-PLC activity, could be dependent on an increase in the substrate PIP_2_ levels, as observed in response to abiotic stress in plants [13].

### 3.2. Inositol 1,2,3,4,5,6-Hexakisphosphate as a Putative Signaling Mediator

Myo-inositol-1,2,3,4,5,6 hexaskisphosphate (IP_6_ or phytic acid) is a component of plant cells that regulates many cellular functions. In plants, IP_3_ might be phosphorylated into IP_6_ by two inositol kinases, inositol polyphophate multikinase 6/3 (IPK2), and inositol polyphosphate (IPK1). IP_6_ accumulates in large amounts in seeds, pollen, and other storage tissues, where it serves as a source for Pi, inositol, and minerals [3,24]. As a signaling molecule, IP_6_ has received attention in recent years. Some authors point out that IP_6_ is the central signaling molecule rather than IP_3_ [25,26,27,28]; however, it is also clear that there is an important contribution of IP_3_ as a precursor for IP_6_ generation. In contrast, there are reports showing that IP_6_ controls cellular reactions through the mobilization of intracellular Ca^2+^ deposits. For example, Lemtiri-Chlieh et al. [25] suggested a signaling role of IP_6_ in abscisic acid (ABA)-regulated Ca^2+^ release in guard cells in which the vacuole may contribute to the release of Ca^2+^ in response to IP_6_. In this way, it is necessary to determine whether these molecules send different signals in plants, and it would be interesting to undertake studies that allow evaluation of the impact of IP_3_ and IP_6_ on the same cellular response.

### 3.3. Phosphatidic Acid

Phosphatidic acid (PA) may be formed from structural membrane lipids such as (PC and PE by phospholipase D, mainly to produce PA species such as PA 18:3/18:2 and PA 18:2/18:2. Additionally, the combined action of PI-PLC and diacylglycerol kinase (DGK) generates the PA species 16:0/18:2 and 16:0/18:3 [29]. Therefore, lipidomic tools have allowed research to reveal which metabolic pathway is activated in response to stress. Differential ^32^P radiolabeling and chromatography technique has been most commonly used to reveal the signaling mechanisms that are involved in hormonal signaling, cytoskeleton, and vesicle trafficking [30,31,32,33,34]. One limitation biochemistry methodologies have faced is that cellular levels of PA are highly dynamic in response to stimuli and to the various enzymatic reactions that modulate its production and degradation.

The role of PA, as a second messenger, has been established by identifying PA-binding domains (PABD) within PA effectors in different plant cell processes. This suggests the importance of this molecule as a central messenger in phospholipid-mediated signaling. Recently, an increasing number of PABDs fused with fluorescent proteins have been used as probes to obtain images of the spatiotemporal distribution of PA in plant cells [35,36]. For instance, the PABD-derived probe Spo20p (Spo20p-PABD) was fused with YFP to monitor PA in growing pollen tubes in tobacco [35]. This biosensor allowed us to detect that the different distribution of PA in the subapical zone is important in the regulation of endocytosis and in the actin dynamics for growth of the pollen tube. Using an optogenetic biosensor, Li et al. [36] development a probe with NADPH oxidase PA-binding domain (RBOHD-PABD) based on Förster resonance energy transfer (FRET) and found that biosensor can monitor the dynamic changes in PA in the plasma membrane in Arabidopsis cells in response to saline and hormonal stress. These findings have contributed to understanding the dynamics of PA in cells under specific environmental conditions, however there is still the challenge of delving into the subcellular distribution of PABD when expressed as PA sensors fused with XFP in response to stress.

Another aspect that has been addressed for the study of PA is through the enzyme PLD. Genetically modified plants have also been used to address the role of some PLD isoforms in the production of PA in response to abiotic stress [37,38]. The results showed that the cellular response derived from the activation of the PI-PLC pathway is functionally different from that resulting from PLD, although both enzymes can generate PA.

For a thorough understanding of the molecular mechanism by which PA regulates different developmental processes in plants, the reader is referred to many excellent reviews on this subject [39,40,41,42].

### 3.4. Other Lipid Second Messengers

The roles of other lipid classes in plant cells during abiotic stress, such as sphingolipids and lysophospholipids, have recently been discovered. The term sphingolipids covers a class of lipids composed of the following three blocks: the long chain base (LCB), the amide-linked fatty acyl chain to the LCB, and the polar head group. LCB is considered the simplest functional sphingolipid and may be linked to a very-long-chain fatty acid via an amide bond to form a ceramide [43]. LCB esterification with a phosphate group at C1 occurs to form phosphorylated LCBs (LCB-P). In plants, the different classes of sphingolipids and LCB-Ps allow these molecules to function both as bioactive lipid components to regulate diverse cellular processes, including signaling, and as structural components in the membrane in plant cells [43,44]. Although the first evidence of the role of LCBs as second messengers was reported for stomatal closure [45,46], its identity remains unclear. For this reason, several research groups have focused on genetic analysis with mutants to establish whether a particular LCB-P is a mediator of signaling. Michaelson et al. [47] analyzed a mutant with a T-DNA insertion in the *4-desaturase* gene in Arabidopsis and exposed it to ABA. Their results showed that phosphorylated 4E-sphingenine (SPH-P) was not involved in stomatal closure in Arabidopsis. In contrast, complex sphingolipids such as glucosyl ceramide (GlcCer) and glucosyl inositol phosphoryl ceramides (GIPC) have also been reported in plant tissues; however, they have not yet been assigned a role as signaling molecules in plants. Thus, an interesting question to be investigated is whether plants possess an enzymatic degradation pathway for structural and complex sphingolipids such as GIPCs to generate signaling molecules involved in the response to stress in plants. For more details on sphingolipid biosynthesis, see the recent reviews by Huby et al. [43] and Cassim et al. [48].

Lysophospholipids (LPLs) are phospholipids that harbor one fatty acyl chain and are generally produced from a large pool of glycerol- and sphingosine-based phospholipids in the membrane lipid bilayer by phospholipase A [1]. Examples of these are lysophosphatidic acid (LPA), lysophostatidylcholine (LPC), sphingosylphosphorylcholine (SPC), and sphingosine 1-phosphate (S1P). The signal functions of LPLs are much less well documented than those of phospholipids. For instance, LPA has been suggested to participate in osmotic signaling in algae [49]. LPC and S1P, have also been proposed as second messengers in plant cells [50,51]. In 2007, Drissner and coworkers reported that LPC is an important signal in arbuscular mycorrhizal symbiosis in *Solanum tuberosum* L.

These findings infer that LPLs exhibit a wide range of biological activities. It is therefore necessary to elucidate the underlying mechanisms by which the LPLs signal is transduced in plant cells. One aspect that has been addressed is the identification of receptors. Although in animal cells it has been established that the effect of LPLs is mediated by G protein-coupled receptors (GPCRs), this in plants is still controversial. Coursol et al. [52] reported that heterotrimeric G proteins have been identified as molecular elements in S1P signaling during ABA regulation in Arabidopsis guard cells. In contrast, Wielandt et al. [53] reported that plasma membrane ^+^H-ATPase (PM ^+^H-ATPase) as a lysophospholipid receptor evidenced the participation of LPLs as important plant signaling molecules in the regulation of electrochemical gradients in Arabidopsis. 

## 4. Link between Lipid Second Messengers and Osmotic Stress

### 4.1. Osmotic Stress-Induced Lipid Second Messengers

Osmotic stress is one of the most important abiotic stresses for crop productivity. Plant cells experience osmotic stress when the solute concentrations in their apoplast change and respond with compensatory adaptations to reestablish osmotic equilibrium [4]. To survive osmotic stress, such as high salinity or dehydration, plant cells activate signaling pathways that lead to a wide range of responses in gene expression and metabolism. Although the importance of salinity and drought has been recognized for a long time, the identity of the molecular components involved in signaling tolerance in plants has been gradually established. Evidence has shown the importance of lipid-mediated reorganization of cell membranes, as well as its role in signaling to respond to changes in osmotic stress in plant cells [54,55,56]. However, more work is needed to fully describe the impact that lipid second messengers have on the molecular landscape during osmotic stress in plant.

### 4.2. IP_3_ and IP_6_ upon Osmotic Stress

A worldwide problem in the cultivation of plants is caused by high salinity in soils, which causes cells to lose water and experience reduced turgor pressure [57]. Osmotic stress imposed by NaCl or KCl generates a rapid increase in IP_3_ and mobilization of Ca^2+^ in different models of plants, such as Arabidopsis [58,59], *Daucus carota* L. [56,60], and *Nicotiana tabacum* [15]. Previous work has reported that osmotic stress activates the PI-PLC pathway [61]. For example, Hirayama et al. [62] reported a PLC gene, AtPLC, in Arabidopsis that is induced by salt and drought stress. Another study [61] analyzed the expression patterns of TaPLC1 under drought and high salinity stress (200 mM NaCl or 20% (*w*/*v*) PEG) in wheat plants. Their data showed that the expression of TAPLC1 was low in the seedling stage and was strongly induced under osmotic stress conditions. Additionally, in our group, Usorach (2016) (unpublished data) observed a 20% increase in the in vitro activity of PI-PLC by ^3^[H]-IP_3_ formation in barley coleoptiles under conditions of saline stress (NaCl: 50200 nM), which was contrary to that observed by osmotic stress with mannitol and sorbitol (100–400 nM). Interestingly, in barley roots, PI-PLC activity increased by 50% under both saline and osmotic stress (unpublished data).

These results suggest that PI-PLC activity is different for each plant tissue that is subjected to osmotic stress, though it must also be take considered that enzymatic activities are affected in plants by the type of stress.

Additionally, the use of pharmacological approaches, such as PI-PLC inhibitors, has provided a molecular view of the link between the PI-PLC pathway and IP_3_ under osmotic stress. This strategy has made it possible to observe how the calcium signal is affected by inhibiting the production of IP_3_ and blocking metabolite biosynthesis induced by water stress. In this context, Parre et al. [63] reported that the inhibition of PI-PLC by U73122 decreased IP_3_ levels and in the Ca^2+^ signaling. These results showed that Ca^2+^/PI-PLC signaling is a committed step in the biosynthesis of proline (an osmolyte) in response to water stress. Recently, a connection between phosphoinositides and osmotic stress gene expression was also demonstrated. Takahashi et al. [59] reported that hyperosmotic stress induces a rapid and transient elevation in IP_3_ levels in Arabidopsis T87 cells due to PI-PLC activation. However, when the cells were treated with neomycin and U73122, not only the levels of IP_3_ but also the expression of hyperosmotic stress-inducible genes decreased under hyperosmolality.

The involvement of IP_3_ as a lipid second messenger is still controversial because the increase in the levels of IP_3_ contrasts with the relatively high levels of IP_6_, which consequently generates a potent release of Ca^2+^ compared to IP_3_. The two explanations for this could be: (1) IP_6_ is also an important form of phosphate storage (e.g., seeds), so tissue specificity is an important factor for that response; and (2) the constant breakdown of inositol polyphosphates (IPPs) causes a flux from IP_6_ and consequently Ca^2+^ release. However, there is still a long way to go to clarify the IP_6_ signaling mechanism in plants [3,64].

Guard cells, as an experimental model, have made it possible to study the role of IP_6_ in the ABA (drought stress hormone) response, which induces stomatal closure, conserving water and ensuring plant survival [65]. In an interesting work, Lemtiri-Chlieh et al. [25] demonstrated by laser scanning confocal microscopy in dye-loaded patch-clamped guard cell protoplasts that the detected increase in cytoplasmic Ca^2+^ was due to its release from endomembrane stores triggered by IP_6_.

In contrast, signaling PIs are terminated through the action of PI phosphatases and inositol polyphosphate phosphatases (PTases). In the case of IP_3_, 5TPases have the ability to hydrolyze it to prevent its accumulation and consequently alter the oscillations of Ca^2+^ in stress-related pathways. In this sense, strategies such as mutation or overexpression of inositol type I 5PTase genes have been used to establish the importance of IP_3_ in saline signaling. For example, Golani and coworkers [66] reported that T-DNA insertion mutants of At5PTase9 increase salt sensitivity and that overexpression of this gene increased salt tolerance. Another example is Arabidopsis SAL1 [also known as FIERY1 (FRY1)], a gene encoding an inositol polyphosphate-1 phosphatase in Arabidopsis that enhances salt tolerance.

Multiple laboratories have developed mutants to evaluate the importance of FRY1 in of IP_3_ metabolism [55,67,68]. It has been shown that *fry1*-mutant plants treated with ABA induces a sustained increase in IP_3_ levels (not transient levels), which improves stress responses. For instance, Xiong et al. [55] showed that loss-of-function mutations in FRY1 enhanced the induction of stress-responsive genes such as RD29A, KIN1, and COR15A upon drought, salt and ABA treatments. However, overexpression or ectopic expression of Arabidopsis SAL1 could not enhance salt tolerance [69]. These findings are very interesting and have allowed us to raise the possibility that specific genes could be regulated through a different pathway.

### 4.3. Involvement of PLD-Derived PA in Osmotic Stress

PA plays an important and complex role in plant drought and salt stress tolerance in plants [70]. PA reportedly has the ability to act as a docking site for proteins that play an important role in salinity or drought conditions. Likewise, putative proteins and PA binding motifs have been identified, making it possible to know the identity of the signaling components involved in the response to osmotic stress [41,57]. In this context, McLoughlin et al. [41] identified eight putative PA-binding proteins recruited to membranes in response to salt stress in Arabidopsis roots through a proteomic approach. Among these were clathrin heavy chain (CHC) isoforms and glyceraldehyde 3-phosphate dehydrogenase (GAPDH), which were recruited towards the membrane for their interaction with PA in response to saline stress. Other examples are proteins of the sucrose nonfermenting-1-related protein kinase 2 (SnRK2) family [29,41,71]. Julkowska et al. [57] performed an in planta study to characterize the interaction of the PABD in SnRK2 upon saline stress. Their results showed that PABD/domain 1 in SnRK2.4 plays a role in the response to saline stress in Arabidopsis. An interesting approach was taken by Yu et al. [72] to investigate the relationship between PA and MAPK (mitogen-activated protein kinase) signaling in response to salt stress in Arabidopsis. The authors reported that salt stress induces a transient increase in the amount of PA and its binding to mitogen protein kinase 6 (MPK6) and stimulates its kinase activity, which phosphorylated salt overly sensitive 1 (SOS1 Na^+^⁄H^+^ antiporter) [72]. However, knockout of PLDaœ1-derived PA resulted in the generation of less PA and reduced MPK6 activity, leading to the accumulation of more Na^+^ in leaves and increased sensitivity to NaCl stress.

In contrast, some reports have that the molecular species of PA (i.e., PAs with different fatty acyl chains) may exhibit different affinities towards their target proteins [73]. For example, the PA molecular species 16:0/18:2 has the highest affinity for MAPK6. Together, these results indicate that the regulation of PA towards its target proteins under stress conditions is extremely complex due to (1) the fatty acid composition of PA formed by the different contributions of the PI-PLC and PLD pathways that would be active, (2) different PA species interacting with the different target proteins, and (3) different isoforms of PLD and their preference for different substrates (i.e., PC, PE, or PG). This raises the possibility of specificity in signaling, which consequently allows interaction with different effectors. PA and cytoskeletal dynamics are intimately interconnected in plant cells to adapt to saline concentrations. During the response to salt stress, plant cells undergo microtubule depolymerization and reorganization, and both processes are believed to be essential for plant survival under salt stress [64,74]. However, what are the molecular mechanisms that mediate the changes in actin or tubulin dynamics by PA?

Currently, it is known that both the enzyme PLD and its PA product are important regulators of the behavior of actin filaments through the regulation of actin capping proteins (CPs) [75] or the arrangement of cortical microtubules [76,77]. In this context, it has been reported that PLD may be a linker that connects microtubules with the plasma membrane. Gardiner et al. [78] reported a microtubule-binding protein (MAP) with PLD activity in Arabidopsis. Later, Lee et al. [79] demonstrated for the first time that PLD is involved in the regulation of the actin cytoskeleton in soybean cell culture, since the exogenous addition of PA induced actin polymerization. Through proteomic analysis, it has been observed that tubulin is also a target protein for PA [29] and that the binding might not be direct but might occur through a MAP called AtMAP65-1, since the increase in PA by PLDα action recruits AtMAP65-1 to the membrane and induces stabilization of the microtubules, which confers survival against saline stress [80]. Our understanding of the role of PA formation in the osmotic stress response has greatly increased through traditional model systems. However, it is necessary to explore the potential mechanisms by which PA causes downstream effects in emergent models that are of agronomic importance. For example, barley (*Hordeum vulgare*) crops are severely affected by the high salinity of soils and hyperosmotic stress, which makes them excellent experimental models to study the role of PA and its relationship with tubulin in the cytoskeleton. Probing this hypothesis, we explored by confocal microscopy whether microtubule organization was affected by osmotic stress, when the coleoptiles and barley roots were treated with NaCl and mannitol (Figure 2A–F). The distribution of the microtubules was heterogeneous in the cytoplasm of the coleoptiles cells subjected to saline stress (Figure 2B) while in the roots, it became evident that the organization of the microtubules was interrupted by the increase in intracellular compartments (Figure 2E) compared to the control. In relation to mannitol, no differences were observed in the distribution of microtubules in coleoptiles and roots (Figure 2C,F). These results indicate that the activation of PLD under saline stress is important for the reorganization of microtubules in coleoptiles and barley roots, but whether PLD interacts directly or indirectly via PA needs to be determinate.

### 4.4. Other Second Messengers Involved in Osmotic Stress

In comparison to the large body of work related to sphingolipids in the mammalian system, there is a paucity of published studies analyzing bioactive sphingolipids in plants [81]. However, an understanding of the roles of sphingolipids in the response to osmotic stress has been facilitated by mutants in plants. Experiments by Wu et al. [82] showed that AtACER (an *Arabidposis thaliana* alkaline ceramidase) mutants were more sensitive to salinity stress and displayed increased ceramides and reduced LCBs, which suggests that ceramides are an important component in the response to salinity. In another work, Zhang et al. [80] explored the participation of the rice S1P lyase gene (OsPL1) in transgenic tobacco plants under saline stress through a functional analysis. Another bioactive component that has generated interest is sphingosine kinase (SPHK), which phosphorylates phytosphingosine to generate phyto-S1P. This is due to its interaction with PA as a component in transduction and the ABA effect in stomatal closure. Guo et al. [83] reported that SPHK1 and SPHK2 are molecular targets of PA and are part of the signaling networks in Arabidopsis. This could suggest that the interaction between PA and sphingolipids is a critical point to coordinate the response to stress in plants. Another compound analyzed is phosphoryl ceramide (GIPC). Jian et al. [84], using a mutant, identified the importance of plant-specific GIPC sphingolipids in the modulation of salt-associated ionic stress in the plasma membrane. Recently, Yang et al. reported that NaCl (300 nM) inhibited sphingolipid accumulation in a ceramide kinase-deficient mutant. These observations suggest that these compounds may also fulfill important signaling roles. Although there is no direct evidence linking sphingolipids and salt stress, sphingolipidomic analysis could yet reveal a link.

In relation to lysophospholipids, it has been suggested that LPC could be a candidate second messenger since it regulates different protein kinases, phosphatases and other signaling molecules. For example, MPK6 has been reported to be a target protein for lysophospholipids derived from pPLAIIIγ. Studies suggest that the activation of MAPK6 causes the phosphorylation of the antiporter Na^+^/H^+^ SOS1, which contributes to reducing Na^+^ levels in plants [85]. In contrast, analysis of a pPLAIIIγa knockout mutant in Arabidopsis showed that the plants were sensitive, while overexpression improved the tolerance of the plants to saline stress [86]. Future studies should be carried out to establish whether pPLAIIIγ responds by modulating other independent pathways to SOS during osmotic stress.

## 5. Conclusions and Perspectives

Plants constantly face different types of abiotic stresses and their response involves the generation of second messengers. In this review, we summarize the second messengers derived from lipids and the molecular scenarios of their involvement in the response to osmotic stress in plants (Figure 3). Interestingly, multiple studies indicate that these second messengers drive downstream responses involving protein-protein interactions. Although research using omics studies has contributed to the understanding of the mechanism that these signaling molecules carry out, it is necessary to further exploit the field of genetic manipulation. In this sense, it would be interesting to use editing technologies and genetic approaches such as knockout lines, to learn more about the function of IP_6_, lysophospholipids and sphingolipids in planta in other experimental models. Another aspect to be addressed is the identification of more molecular targets of lysophospholipids and sphingolipids that allow to explain the effects of osmotic stress in different plant cells. Therefore, in the future, efforts should be devoted to conducting new studies that combine genetic and molecular approaches that could contribute to the understanding of osmotic signaling in plant cells. In conclusion, lipid second messengers are important players in osmotic signaling in plant cells, and there are still potential studies that need to be conducted to clarify the molecular mechanism. This will allow to development of strategies to generate crops with least negative impacts on normal physiology due to osmotic stress.

## Figures and Tables

**Figure 1 ijms-22-02658-f001:**
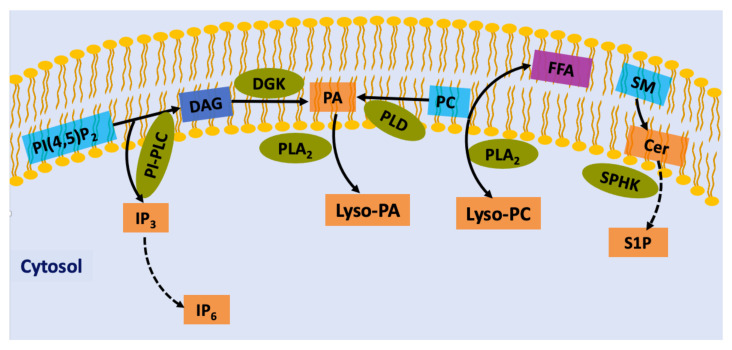
Generation of lipid second messengers in plants. Phospholipid precursors (blue box) involved in the production of intracellular second messengers (orange box). PI-PLC leads to the cleavage of PIP_2_ into DAG and IP_3_. DGK converts DAG to PA, which is a second messenger on its own right. PA, which can also be generated by PC hydrolysis by PLD. DAG can also be synthesized from PC via PLD. IP_3_ diffuses into the cytosol, where it is converted to IP_6_. Fatty acids of phospholipids are liberated by PLA2s and converted to eicosanoids. Lysophospholipids are also precursors of a different class of lipid mediators, including Lyso-PC or Lyso-PA. Sphingomyelin is a precursor of ceramide that can then be phosphorylated to generate ceramide 1-phosphate and to form sphingosine, which is phosphorylated to generate sphingosine 1-phosphate. PIP_2_, phosphatidylinositol (4,5)-bisphosphate; PC, phosphatidylcholine; PI-PLC, phosphoinositide-phospholipase C; IP_3_, inositol (1,4,5)-trisphosphate; IP_6_, myo-inositol-1,2,3,4,5,6 hexaskisphosphate; DAG, diacylglycerol; PLD, phospholipase D; PA, phosphatidic acid; PLA2, phospholipase A2; Lyso-PA, lyso-phosphatidic acid; Lyso-PC, lyso-phosphatidylcholine; FFA, Free Fatty Acid; SM, sphingomyelin; Cer, ceramide; SPHK, sphingosine kinase; S1P, sphingosine-1-phosphate.

**Figure 2 ijms-22-02658-f002:**
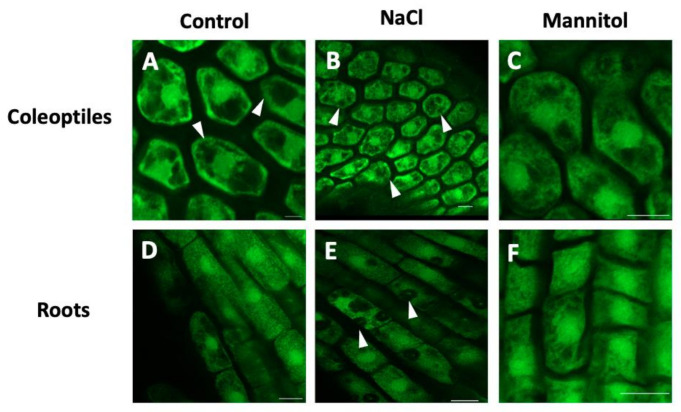
Organization of microtubules in coleoptiles and barley roots under osmotic stress. The images (**A**,**D**) show the distribution of the microtubules in the central plane of the cells of the coleoptile apex and the radical apex in roots without treatment. The distribution of the microtubules was disrupted when cells of coleoptiles and root were treated with NaCl (100 mM, images **B**,**E**) or mannitol (200 mM, images **C**,**F**). Fluorescence-labeled microtubules were visualized with a confocal laser microscope (Nikon Eclipse Ti). Scale bar = 20 µm.

**Figure 3 ijms-22-02658-f003:**
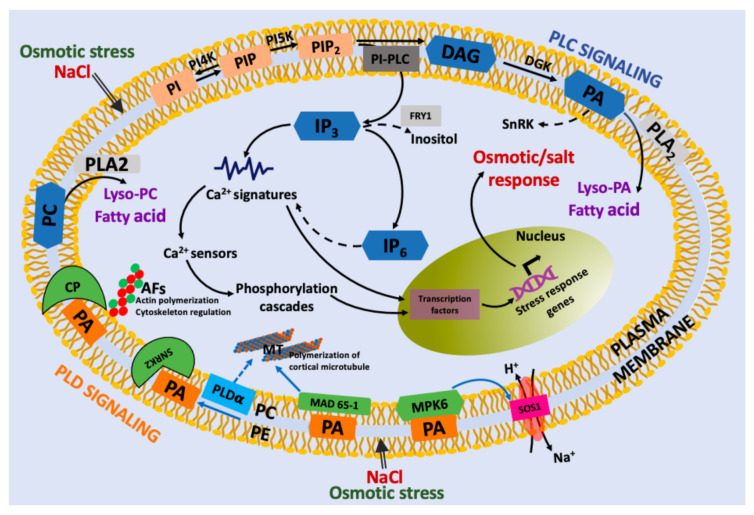
Proposed model for lipid-derived second messengers under osmotic and salt stress in plant cells. Osmotic and salt stress is perceived at the cell membrane, which activates PI-PLC, PLD and sphingolipid signaling to produce lipid second messengers that trigger the release of calcium from different sources, directly or indirectly. The changes in calcium concentration are sensed by calcium sensor proteins (e.g., CaM calmodulin, CML calmodulin-like protein sensors). In this response, PI-PLC and PLD signaling promotes a chain of reactions that includes IP_3_, IP_6_, and PA. PA has numerous targets, such as SNRK2 (snf1-related protein kinase2), MAD 65-1 (microtubule-associated protein MAD 65-1), and MPK6, that produce diverse cellular effects, such as actin and cortical microtubule polymerization. Finally, lysophosphatidic acid (lyso-PA) or lysophosphatidylcholine (LPC) can also stimulate many cellular processes.

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
