# Peer review of "Link between Lipid Second Messengers and Osmotic Stress in Plants"

_ijms, 2021, doi:10.3390/ijms22052658_

Round 1

Reviewer 1 Report

This manuscript by Rodas-Junco is intended as an overview of phospholipid signaling, which is necessary and timely. However, it fails at being useful in its current form for several reasons. In part, this is due to sometimes incorrect phrasing; in part because the authors use PI, PIP and PIP2 sometimes as synonyms, which they are not. Third, the manuscript is also often quite superficial. While this may be the point of a review, a bit more depth or a few more specific discussions would greatly improve it. For example in line 370 the authors write. “A study conducted by Guo et al. [79] reported that SPHK1 and SPHK2 are molecular targets of PA and are part of the signaling networks in Arabidopsis.” This is not a useful statement and the authors should at least bother to list which signaling pathway and which stress response is affected.

In general, the manuscript is not coherent. While the various categories make sense, findings are often not listed in logical order within the paragraphs. The authors should rearrange the review to flow more logically.

Abstract:

L.17-20 The authors write “In this sense, it has been shown in various plant models that

membrane lipids are substrates for the generation of second lipid messengers such as phosphoinositide’s, phosphatidic acid, sphingolipids and lysophospholipids.”  

L.19: It needs to be “phosphoinositides” not “phosphoinositide’s”

Introduction:

The authors group salt and drought together under osmotic stress. While this is somewhat justifiable, salt and drought elicit responses that are specific for the stress. For this reason, the authors should not groups them together but at least point out, that they are distinct stresses with distinct responses.

  1. 50. The authors write ” In this sense, membrane lipids can be converted by modifying such enzymes as phospholipases….into signaling molecules known as lipid-derived second messengers.” NO. This is incorrect English. It is not the lipid that modified the lipase to generate a signaling molecule but the lipase that modifies/processes the lipid to generate the signaling molecule.

Figure 1: While I like the concept of the figure, it needs to be modified: The enzymes should be drawn on the inside of the cell since they are soluble and not membrane proteins. Also, I would recommend to only put the metabolites/ lipids in square boxes and the enzymes in ovals (and not at endpoints of arrows)

L.72++ The authors should consider using PI, PIP and PIP2 to distinguish between the three different lipid classes as this becomes very important (and otherwise confusing) in the subsequent parts of that paragraph. The way this part of the review is written using PI and PIP2 at times as synonyms and other times to indicate distinct lipids, will annoy anyone familiar with phospholipids and confuse any reader new to the field. Either way, they will likely stop reading by page 3.

L.95+ This sentence would benefit from a better connection to the previous sentence. Also, PLD “can” produce PA is a bit misleading. To my knowledge it “does” produce PA. PI3K should be described before the lipases as it generates PIPs rather than cleave them.

Why do so many sentences start with “In this sense,….”? This is unnecessary.

The font keeps switching. L. 103++ uses bold for regular text. It starts with a statement about the search for a IP3 receptor, but rather than discussing this further, it then lists a bunch of random findings/responses that lead to IP3 production. This order that makes no sense. The conclusion statement is actually fine if there wasn’t the disconnected detour to the receptor earlier in this paragraph.

L.128 ++ is nicely structured and described.

L.143++ this is structured well also. It is very superficial, though, and could go into more depth. The reviews that are cited are from 2013 and 2018. A lot has happened since then and should be reported here.

L.162+++ Do the authors really think their description helps in any way in understanding what a sphingolipid is?

They are making repetitive statements:” Although the first evidence of the role of LCBs as second messengers was reported for stomatal closure [43, 44], its identity remains unclear. For example, the role of LCBs in stomatal signaling remains unclear. In this sense,…” This can be shortened. And followed with “For this reason,….”

L.196; The authors are aware that research are now looking at the role of heterotrimeric G-proteins in plants, including their role interacting with lipids/lipid signaling???

L.207++ The authors write “Evidence has shown the importance of lipid-mediated reorganization of cell membranes as well as its role in signaling to respond to changes in osmotic stress in plant cells [51-53]. However, the mechanism by which this type of stress activates phosphoinositide signaling is not yet clear.” Maybe it would be clearer if the authors read reference that are younger than 2015? I was missing a lot of the references on PA and stress signaling here but they are coming later at line 276+++

L.232: The authors correctly conclude from their figure 2 that “These results infer that the activity of PI-PLC is different for each plant tissue that is subjected to osmotic stress.” They should also point out that the type of stress affects the enzyme activity as well.

The authors discuss the possible signaling function of IP6. However, it is also considered a phosphate-storage molecule. This would explain its high abundance in at least some tissues and should be at least mentioned.

L.276++ this paragraph is a summary of the role of PA in signaling. It is a bit narrow in that it focuses on the interaction with microtubules but written very nicely. However, the findings the authors report in figure 3 do not appear to be significant and are not reported correctly.

Reviewer 2 Report

On the basis of my careful consideration of the manuscript, I found this work very interesting in terms of reviewing the recent advances of the link between lipid second messengers and osmotic stress in plants. Overall, the manuscript is well written and in a logical order. I feel that this nice review will be interesting not only to scientists working with this field but also to a broad scientific audience.

Reviewer 3 Report

Paper "Link between lipid second messengers and osmotic stress in plants" is interesting and very important but needs some corrections.

Figure 2 needs LSD or HSD values.

Figure 3G needs LSD or HSD value.

Figure 3H needs LSD or HSD value.

In Review type papers very important is a meta-analysis. In this paper lack of meta-analysis.

Paper needs major revision.

Round 2

Reviewer 3 Report

Now, all is perfect.